# Calculation and Experimental Study of Low-Cycle Fatigue of Gas Turbine Engines Booster Drum

**Alexander Arkhipov \*, Yury Ravikovich, Dmitry Kholobtsev and Alexander Shakhov**

Moscow Aviation Institute (National Research University), 4 Volokolamskoe Shosse, Moscow 125993, Russia; yurav@mai.ru (Y.R.); dima67@list.ru (D.K.); shakhov_alexander@mail.ru (A.S.)

\* Correspondence: arkhipov.48@list.ru

**Abstract:** A calculation-experimental study of durability of titanium booster drum of gas turbine engine (GTE) was carried out. A methodology of experimental estimation of titanium component life of GTE using structurally similar elements (SSE) has been proposed. A series of three-dimensional calculations by the finite element method of SSE cut out of the finished part was carried out to estimate the strength of the booster drum. A methodology for testing the durability of SSE in the low-cycle fatigue (LCF) area was developed, and statistical processing of the test results was performed. Tests of SSE allowed carrying out advanced-edge assessment of the durability of a full-size drum, taking into account the manufacturing technology.

**Keywords:** gas turbine engine; finite element method; low-cycle fatigue; stress–strain state; structurally similar element

## 1. Introduction

An important task in the design of modern aviation GTE is the need to provide a service life of more than 20,000 cycles, while reducing the specific weight, fuel consumption and increasing the specific thrust. The given problem demands perfection of calculation and experimental methods of maintenance of cyclic durability of rotor engine parts, as well as an increase in accuracy of calculation of stressed-strained condition (SSC) in application of the modern computing complexes based on the use of two-dimensional and three-dimensional finite-element models with the nominal sizes and loadings, experimental check and acknowledgement of reliability of models [1–3].

The most common cause of failure of rotor parts is the appearance of cracks because of cyclic loads, leading to accumulation of fatigue damage in the material of the part, which can lead to destruction [4].

The weak link in assessing the durability of titanium parts is the real LCF curve. Durability under the same stresses can vary significantly depending on the forgings supplied, the forging technology (including heat treatment) and the machining technology.

LCF life zone for titanium alloys over 10,000 cycles is not very studied. Most companies' durability models are based on extrapolation of LCF data in a range of up to $10^4$ cycles or multi-cycle fatigue results in the range of $10^5$ cycles and above [5].

The assessment of durability of parts made of titanium alloy Ti-6Al-4V under cyclic loading is carried out according to the results of tests of standard specimens at different levels of stresses and/or strains [6]. The main ways of determining the durability of the material in SSC zone are shown in [7], and standard specimens also are used. Also, a large number of models described in open publications allow predicting durability only for certain operating parameters [8], from which it can be concluded that each individual GTE part requires an individual assessment of durability. In the present work, an element with a concentrator cut out of a finished product is investigated, which allows taking into account all technological peculiarities of manufacturing.

In order to improve the accuracy of calculations and durability estimation, it is necessary to determine the fatigue characteristics of considered GTE part. One of the main factors influencing the durability of titanium alloys is the geometry and manufacturing technology of the part [9], so durability values can be obtained only by testing this part or its SSE while maintaining the main stress concentrators and considering the manufacturing technology [10], and the influence of cold creep inherent to titanium alloys should be considered [11].

An additional factor limiting the evaluation of LCF characteristics of titanium alloys is the dependence on the frequency of loading and the dwell time at maximum stress. This dependence for titanium alloys is evident even at temperatures close to normal due to "cold" creep.

Thus, the final conclusion about the booster drum durability can be given only after SSC tests.

The main test for critical parts is in-service testing on the engine. In many cases, prototype engine test results are used to verify durability. However, verification of such tests requires a comparative calculation evaluation of durability in operation on the prototype and new engine, carried out taking into account all the differences in component design on the prototype and the new engine and the differences in operating conditions. This type of testing goes on continuously, but the leading estimation of durability is not carried out, and the resource is only confirmed [2].

To assess the service life of rotor parts, additional tests are possible on special test benches. The main type of such tests for military engines are equivalent-cyclic tests of disks in acceleration chambers [2]. However, for civilian engines with a long service life and an integral rotor consisting of several stages, the cost and duration of preparation and conduct of such tests can be unjustifiably high.

This paper describes the methodology of anticipatory life estimation of low-pressure compressor of Power Jet SaM146 engine, produced by venture between Safran S.A. (Paris, France) and UEC Saturn (Rîbinsk, Russia), based on the calculation-experimental method of establishing the service life. For anticipatory estimation of service life, cyclic tests of SSE cut from the finished drum with preservation of the surface in the critical zone are proposed. This paper also demonstrates the methodology of preparation, implementation and analysis of resource tests on a design-like element of a booster drum of a low-pressure compressor of GTE.

## 2. Theoretical Basis

### 2.1. Study Object

Study object is GTE SaM146 [12] for the Sukhoi Superjet 100 (produced by JSC Sukhoi Company, Moskow, Russia) regional jet. The low-pressure compressor of this engine is one of the parts with the lowest design life.

### 2.2. Methodology Brief Description

In this work, the method of finite elements, which is implemented in the software package ANSYS® ANSYS Mechanical (version 2020 R2, CADFEM-cis, Moscow, Russia), is used to determine the stress-strain state. This complex is used in modern aircraft engineering to determine the characteristics of the product at the design stage. These calculations require reliable information about material properties, so to verify our calculations, experimental verification of the validity of the computational model was conducted.

This article describes the steps that are necessary to be able to conduct advanced edge evaluation of the life of GTE booster drum:

- calculation of stress-strain state of the booster drum and determination of dangerous zones;
- development and calculation of SSE cut out of the critical zone, with preservation of the manufacturing technology and the identity of SSC with full-size drums;

- development of test equipment and testing of SSE, confirming the adequacy of the calculation model;
- conducting durability tests of SSE.

Thus, the main task of the work was to develop a methodology for the calculation-experimental study of natural SSE, preserving their manufacturing technology to confirm the life of the product and the possibility of conducting a preliminary assessment of GTE durability.

## 3. Methodology

### 3.1. Calculation of the Booster Drum's SSC in the Critical Zone Area

#### 3.1.1. Application of Geometric Model Parameterization in Strength Calculations

A modern approach to strength calculations requires the preparation of a parametric model of the part, which allows taking into account changes in individual parameters (geometric dimensions and material properties, as well as SSE fixation during tests) and conducts automated rebuilding of the model, which reduces the time for preparation of calculations [5,13–15].

#### 3.1.2. Calculation of the Axisymmetric Rotor Model

At the first stage, we analyzed the axisymmetric model, including the preparation of the parametric model, setting the boundary conditions, transferring the temperature field from the aerodynamic calculation and simplifying the non-axisymmetric elements to unit masses [16]. The axisymmetric calculation makes it possible to obtain a fairly accurate picture of the distribution of displacements (Figure 1) and obtain the boundary conditions for further three-dimensional calculations.

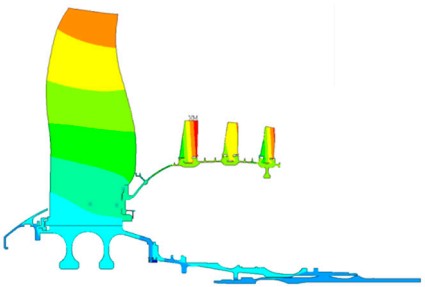

**Figure 1.** Radial displacements of the axisymmetric rotor model.

#### 3.1.3. Calculation of the Three-Dimensional Sector of the Booster Drum

The next stage is to conduct a series of calculations of a three-dimensional sub-model of the rotor sector containing the critical area. A parametric model is also developed for this stage [5]. The calculation takes into account the boundary conditions from the previous axisymmetric calculation, contact interaction of blades and drum, temperatures and centrifugal loads. Stress deviations in the critical zone can be up to 4%, depending on the chosen three-dimensional sector [16]. The Figure 2 shows the distribution of surface stresses in the booster drum sector. This calculation was used for engine certification.

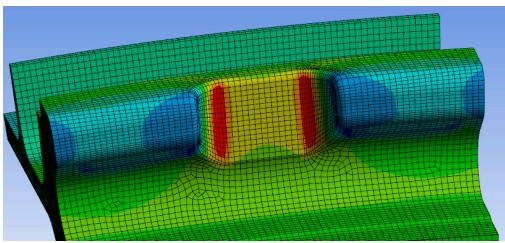

**Figure 2.** Distribution of surface stresses in the critical zone of the booster drum.

Calculations of the booster drum show high accuracy of stress calculation in the critical zone ($\pm 10$ MPa for 5 different finite-element models). On the basis of 2 nominal drum models, 26 models with different deviations of dimensions within the tolerance limits were created. Calculations of these models showed that the difference in stresses for the nominal model and the model with the worst tolerances could be up to 10.8%. It can be assumed that a random (uncorrelated) scatter of tolerances on geometric dimensions of the rotor and the weight of blades with a probability of 0.99% can reduce the maximum stresses [5].

### 3.2. Test Methodology and Development of SSE

3.2.1. Designing SSE and Test Equipment

Servo-hydraulic machines are used for fatigue testing of specimens cut from the rim parts of discs, which are equipped with a fixture for pure bending.

A series of calculations of several variants of SSE was performed to determine the loading force and the subsequent development of the test fixture. For SSE calculations, parametric models created in Siemens Digital Industries Software® Siemens NX were used, which made it possible to carry out a series of calculations in an automated mode. The main criterion for selection was the coincidence of SSC in the critical zone of SSE and the full-size booster drum. The cut-out scheme and general view of SSE are presented in the Figures 3 and 4. The reliability of the developed SSE is confirmed by the coincidence of SSC of the model sample with the certification calculation of GTE booster drum sector and with the results of strain gauges and deflection measurements during subsequent cyclic tests.

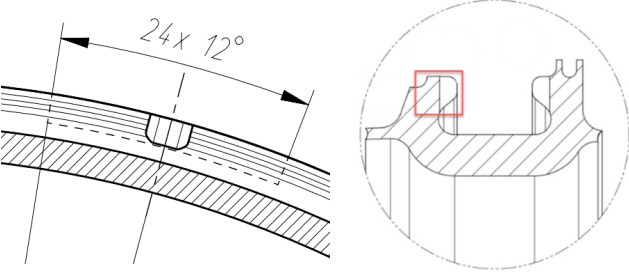

**Figure 3.** SSE cut-out sketch.

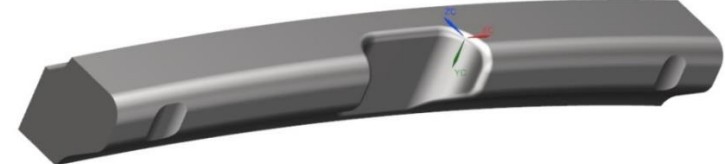

**Figure 4.** General view of SSE cut from the booster drum.

In order to ensure the similarity of SSC with the booster drum, SSE must be loaded according to the scheme (Figure 5), therefore, a special tooling was developed to perform tests on MTS 322 servo-hydraulic machine.

The parametric conjugate model of tooling and SSE is shown in Figure 6a,c; a photograph of the installation is shown in the Figure 6b.

3.2.2. Calculation of SSC of SSE

The coupled parametric model of SSE and test rig has been calculated in the ANSYS software package (version 2020 R2, CADFEM-cis, Moscow, Russia).

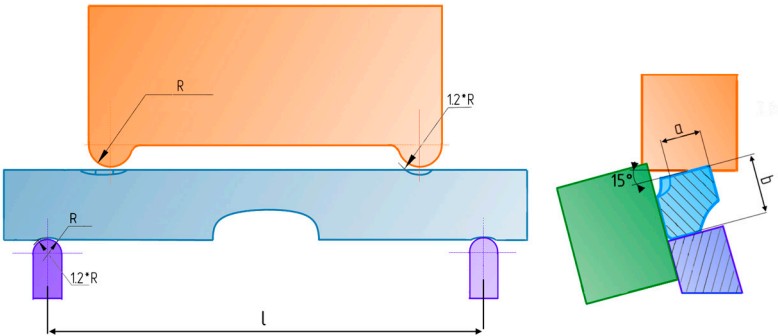

**Figure 5.** Loading scheme during SSE tests at LCF.

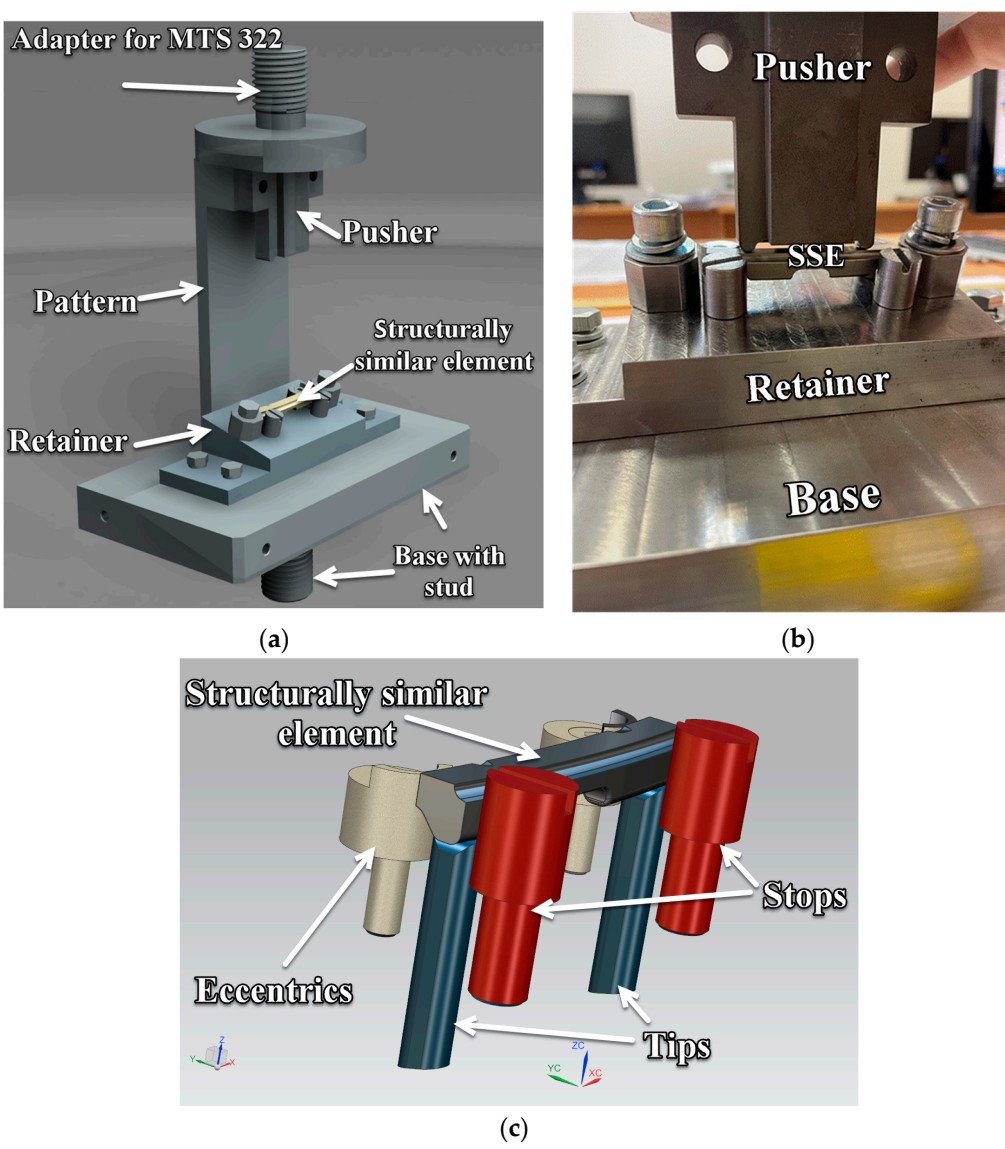

**Figure 6.** Test rig (**a**); matched parametric model (**c**); photo of the manufactured fixture (**b**).

To increase the reliability of modeling, two models of the coupled parametric model of SSE and the test rig were prepared. Calculations of the complete test rig assembly and the simplified model were performed. The finite element grid is shown in the Figure 7a for the full model and in the Figure 7b for the simplified model. The deviations of stress and SSE displacements during the transition from the full model to the simplified model

are insignificant, so further calculations were performed for the simplified model; the characteristics of its finite-element mesh are given in the Table 1.

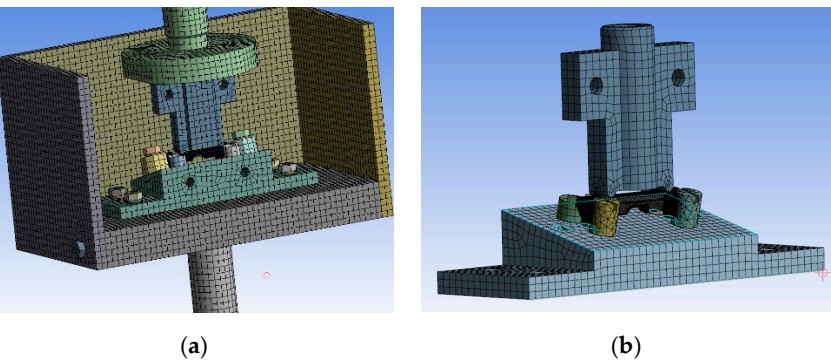

(**a**)　　　　　　　　　　　　　　　　　　　　(**b**)

**Figure 7.** Finite-element grid of the coupled parametric model of SSE and test rig: (**a**) Full model, (**b**) simplified model.

**Table 1.** Characteristics of finite element meshes SSE and test rigging.

| Model | N SSE Units | N SSE Elements | N Equipment Units | N Equipment Elements |
|---|---|---|---|---|
| 1 | 347,532 | 102,563 | 642,507 | 189,642 |
| 2 | 196,934 | 58,831 | 235,647 | 69,553 |

The properties of the materials used in the calculations are shown in the Table 2.

**Table 2.** Properties of materials in the calculation of the coupled parametric model of SSE and test rig.

| Parameter | Value |
|---|---|
| SSE | |
| Density, kg/m$^3$ | 4450 |
| Young's modulus, MPa, | $1.15 \times 10^5$ |
| Poisson's ratio | 0.32 |
| Test equipment | |
| Density, kg/m$^3$ | 7820 |
| Young's modulus, MPa, | $2.06 \times 10^5$ |
| Poisson's ratio | 0.3 |

The boundary conditions for the calculation of the coupled parametric model are shown in the Figure 8. The upper part of the pusher is fixed, a force from the machine drive is applied to the lower part of the retainer and a constraint of displacements along the axes perpendicular to these surfaces is added to compensate for the lack of structural rigidity from the discarded parts on the plane of the retainer, highlighted in blue.

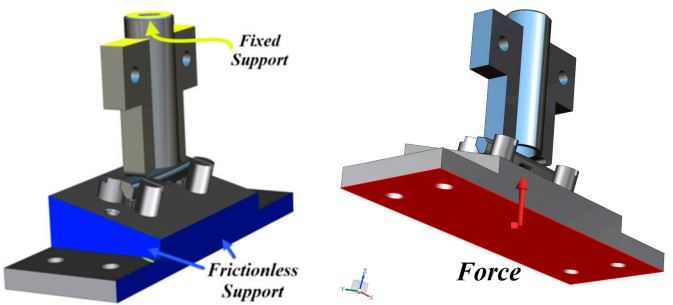

**Figure 8.** Boundary conditions for the calculation of the coupled parametric model.

In contact pairs that include SSE, the contact type is set to frictional with a friction coefficient of 0.05, the parts for which a bolted connection is provided are set to bonded contact.

The bending force is selected under the condition of maximum stress in the critical zone. Based on a series of calculations, an angle of force application $\alpha$ equal to 15° and a force value at which the operating stress in the critical zone is achieved, equal to 4470 N, can be recommended.

With the selected force angle and force magnitude, SSE stress gradient is as close as possible to the booster drum gradient (Figure 9).

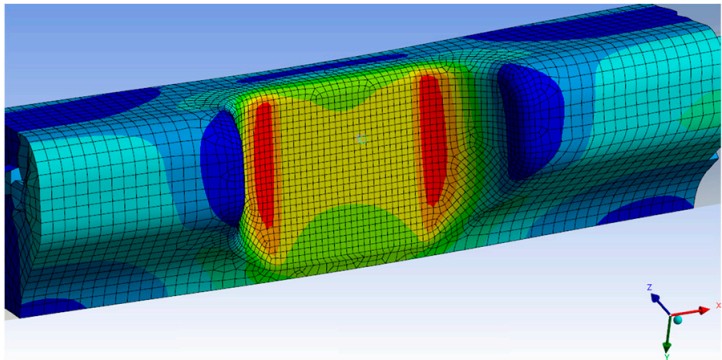

**Figure 9.** Mises stress distribution of the bending sample.

In this case, the difference between the change in stresses along the thickness in the tested specimen and the stresses in the rotating drum with blades is checked [5] (Figure 10).

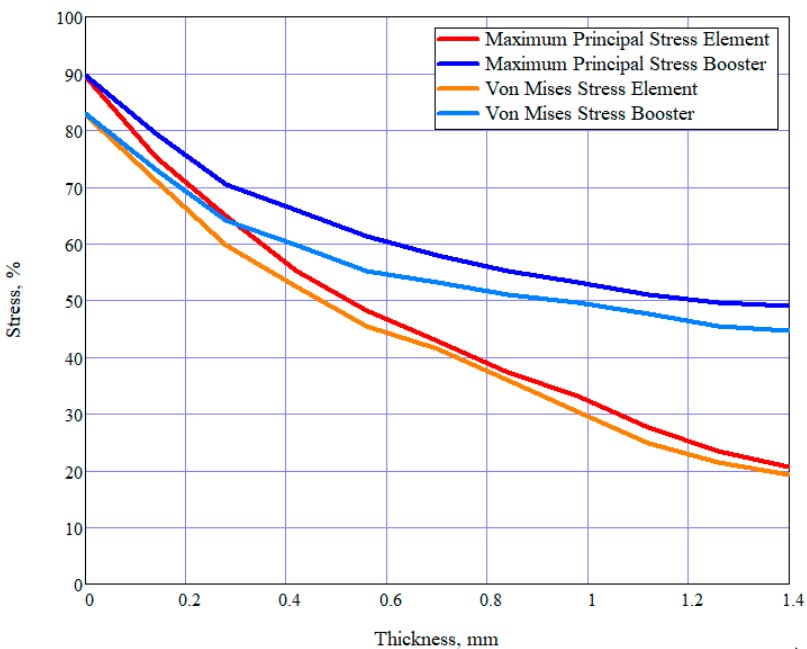

**Figure 10.** Stress gradient in the booster drum and SSE (nominal dimensions).

## 4. Conducting Tests on LCF SSE

Each of the manufactured elements is marked on the front side. For manufactured SSE, the geometry is measured (Figure 11) with the issuance of a measurement passport. The measurements were carried out using two-coordinate measuring device "DIP-3" and lever micrometer "MP 0-25".

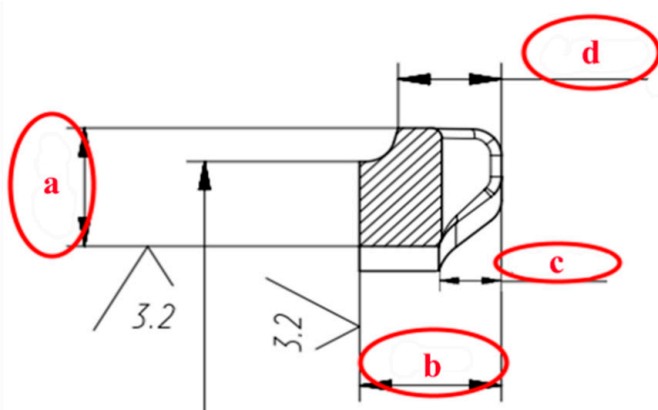

**Figure 11.** Controlled dimensions of SSE.

To control the value of stresses, small strain gauges were used, providing performance at a relative strain of up to 0.5%, and installed according to the sketch shown in the Figure 12. The zone of maximum stresses is at the radius at which strain gauge installation is impossible, so the stress control in the critical zone was carried out on a flat area. Readings of strain gauges at the place of installation were also compared with the calculated values. Load cells were necessary to control the initial loading force. The movement of the rod of the servo-hydraulic machine was also monitored.

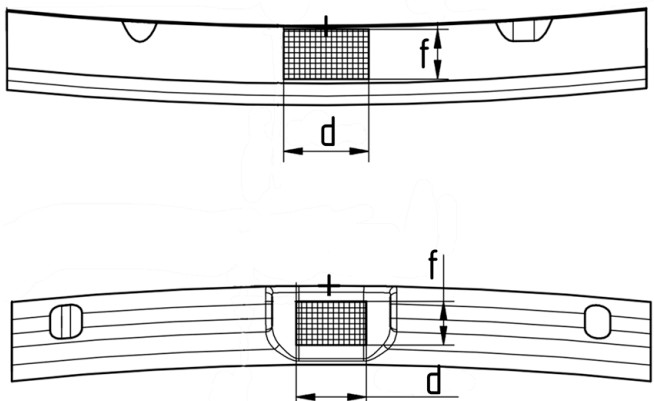

**Figure 12.** Sketch of strain gauge installation on SSE.

To reduce the time, the tests were performed with a pulsating loading cycle with a frequency of 600 cycles per minute (10 Hz) without dwell time at the maximum load. The maximum load, increasing stepwise, was obtained by calculations and monitored with strain gauges at SSE. The minimum residual load was 200 N, the maximum was 4500 N and the calculated stress level was 100%.

The temperature of the test element was controlled by a thermocouple attached in the critical zone. According to the readings, the temperature in the region of the critical zone did not exceed the temperature in the laboratory. Due to the complexity of mounting the thermocouple and the possible impact on SSC, no temperature control was performed for the test series.

The loading and displacement forces were recorded through the test machine control program using on-board sensors and recorded in a test report.

Relative strain values were measured using two strain gauges glued to the specimen at the top and bottom of the cutout. The measurement results are shown in the Table 3.

**Table 3.** Test reports.

| № | Voltage Measured, σ | Load, kN | Number of Cycles, N |
|---|---|---|---|
| 1 | 97% | 4470 | 30,000 |
| | 110% | 5364 | 30,000 |
| | 127% | 6258 | 30,000 |
| | 142% | 7152 | 6425 |
| 2 | **Added to voltage 100%** | | **159,177** |
| | 107% | 4470 | 30,000 |
| | 112% | 5364 | 30,000 |
| | 118% | 6258 | 23,291 |
| 3 | **Added to voltage 100%** | | **134,248** |
| | 96% | 4470 | 30,000 |
| | 105% | 5364 | 30,000 |
| | 132% | 6258 | 30,000 |
| | 140% | 7152 | 4506 |
| 4 | **Added to voltage 100%** | | **141,373** |
| | 101% | 4470 | 30,000 |
| | 115% | 5364 | 30,000 |
| | 126% | 6258 | 30,000 |
| | 142% | 7152 | 26,095 |
| 5 | **Added to voltage 100%** | | **249,985** |
| | 97% | 4470 | 30,000 |
| | 108% | 5364 | 28,185 |
| | **Added to voltage 100%** | | **69,414** |

## 5. Analysis of Results and Discussion

The values of life obtained with stepwise load increases were recalculated to the stress level corresponding to the stress in the critical zone of the booster drum under an unfavorable combination of tolerances. Assuming linear accumulation of LCF damage at different stress levels and based on LCF curve equation in the elastic area $\sigma^m N = C$, where m and C are the constants of the fatigue curve, it is possible to obtain the reduced numbers of cycles.

The recalculation assumed that the fatigue curve of each SSE at the corresponding probability of failure runs parallel to the minimum fatigue curve built for the normal-logarithmic law of distribution of the logarithm of life with a standard deviation SlgN and corresponding to the deviation from the mean values of $-3$ SlgN. The values of the points of the minimum fatigue curve were obtained on the basis of data on the minimum life with a standard deviation SlgN and corresponding to the deviation from the mean values of the titanium alloy T-A6V. The value of the slope index of the fatigue curve was $m = 3.46$. Thus, life $N_i$ of SSE tested at the level of stresses $\sigma_i$, can be recalculated to the initial stress level $\sigma_{HN}$ or any other stress level $\sigma_j$ according to the Equation (1).

$$N_j = 10^{(\log(N_i) + m \cdot (\log(\sigma_j) - \log(\sigma_i)))}. \tag{1}$$

The failure stress level of the specimens was significantly 40–80% higher than the initial test level, and the number of cycles to failure reduced to 100% stress level for each specimen ranged from 69,400 cycles to 250,000 cycles.

The results of a sample of six elements reduced to the same stress level were used to determine the average $\overline{\lg N}$ (N = 157,400 cycles) and the standard deviation SlgN of the sample.

$$\overline{\lg N} = \frac{\sum_{i=1}^{6} \lg N_i}{6} = 5.197, \tag{2}$$

$$N = 157{,}400 \text{ cycles}, \tag{3}$$

$$\mathrm{SlgN} = \sqrt{\frac{\sum_{i=1}^{6}\left(\overline{\lg N} - \lg N_i\right)^2}{6}} = 0.17968369. \tag{4}$$

The small sample size did not allow conducting a standard Pearson agreement test $\chi 2$. Therefore, to check the distribution for normality, the sample of obtained values of the logarithms of life $\lg N_i$ was sorted in ascending order, and for each value $\lg N_i$ in the sample was determined by the probability $p = I/(n + 1)$, where n = 6 is the number of values in the sample and i is the order number of the value from 1 to n.

The graph of the distribution function of the logarithm of life shows (Figure 13) that a linear distribution can be assumed with a high degree of confidence. Then, the probability values p were converted into quantiles of the normal distribution using the inverse function of the standard distribution NORMSINV in EXCEL.

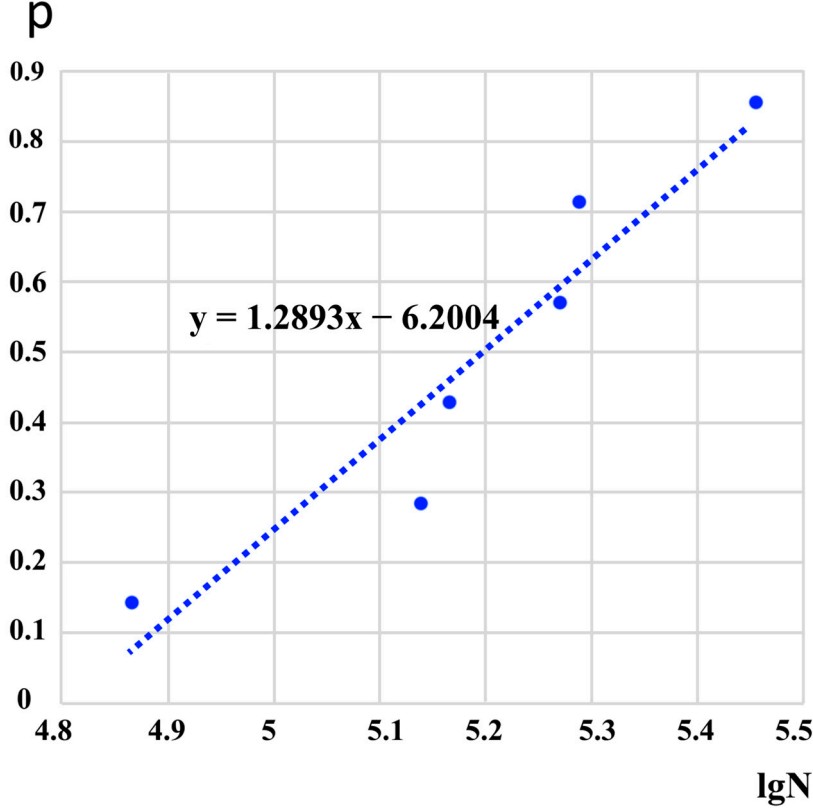

**Figure 13.** Graph of the distribution function of the durability logarithm.

The graph checking the distribution for normality shows that we can assume that the distribution is normal.

## 6. Conclusions

This methodology described in the article can be applied to carry out anticipatory life tests; a reliable value of durability is achieved by taking into account the technology of manufacturing a full-size drum and preserving the stress gradient in the element.

Tests of SSE cut from the investigated drum around the critical zone with preservation of the investigated surface exclude the influence of technological (supplied blanks for forging, forging technology, including heat treatment and machining technology) and design factors on the durability of the drum. The resulting life values will be as close as possible to life y values obtained in cyclic tests of the drum on the engine or on the acceleration test bench. The possibility of testing several SSE allows estimating the service life on a statistical basis.

Six SSE were tested under cyclic loading with a cycle asymmetry ratio of R = 0 with a frequency of 10 Hz and step load increase to failure with an initial stress level of 100%, load increase step of 20% on a test base of 30,000 cycles.

The tests showed that the number of cycles to failure exceeds the specified lifetime of the booster drum of 20,000 cycles.

**Author Contributions:** Conceptualization, Y.R. and A.S.; methodology, D.K.; software, A.A.; validation, A.A., Y.R., D.K. and A.S.; formal analysis, Y.R.; investigation, D.K. and A.S.; resources, A.A. and Y.R.; data curation, D.K.; writing—original draft preparation, A.A.; writing—review and editing, D.K.; visualization, A.A. and A.S.; supervision, D.K.; project administration, A.A.; funding acquisition, Y.R. All authors have read and agreed to the published version of the manuscript.

**Funding:** This research was funded by Ministry of Education and Science (Russia), grant number 075-15-2020-770.

**Institutional Review Board Statement:** Not applicable.

**Informed Consent Statement:** Not applicable.

**Data Availability Statement:** Not applicable.

**Conflicts of Interest:** The authors declare no conflict of interest.

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
