# Peer review of "Calculation and Experimental Study of Low-Cycle Fatigue of Gas Turbine Engines Booster Drum"

_inventions, doi:10.3390/inventions7030049_

Round 1
Reviewer 1 Report
Calculation and Experimental Study of Low-Cycle Fatigue of Gas Turbine Engines Booster Drum, I finished my review and I will share my opinion point, and my decision is to accept after minor revisions. In addition, authors have organized the whole article very well. However, the technical language is also well demonstrated readers can understand very well.
1) Introduction must be improved
1) Line 73 authors are requested to add Power Jet SaM146 readers can understand easily
2) In same line 73 Power Jet SaM146 in brackets requested to put venture between Safran of France and NPO Saturn of Russia. Meanwhile, authors are requested are not hesitate to put highlight there country wherever applicable.
3) In line 78 authors have typo error written that Sam146 it must be SaM146
4) In same line SSJ100 requested to write Sukhoi Superjet 100 and also requested the company designer and origin country in brackets. However, the authors should highlight all the details in the article. Therefore, readers can understand.
5) Ansys Mechanical Software in line 82 requested to add license number, version of Ansys .
6) In same line 82 is Ansys software Ansys® or Ansys Trade mark? any of them add please
7) Figure 3 and Figure 4 can authors are what softwate used by others? if different from Ansys requested to add
8) is there any uncertainty in the work?
9) Figure can Authors represent the same graph in origin, please?
Authors requested to consider my comments and change please. All the best.
Sincerely
Reviewer
Author Response
Reviewer 1
Comment 1: Introduction must be improved.
Answer 1: Introduction was improved.
Comment 2: Line 73 authors are requested to add Power Jet SaM146 readers can understand easily.
Answer 2: “Power Jet” was added.
Comment 3: In same line 73 Power Jet SaM146 in brackets requested to put venture between Safran of France and NPO Saturn of Russia. Meanwhile, authors are requested are not hesitate to put highlight there country wherever applicable.
Answer 3: Information about venture between Safran S.A. (France) and UEC Saturn (Russia) was added.
Comment 4: In line 78 authors have typo error written that Sam146 it must be SaM146.
Answer 4: Typo error has been corrected.
Comment 5: In same line SSJ100 requested to write Sukhoi Superjet 100 and also requested the company designer and origin country in brackets. However, the authors should highlight all the details in the article. Therefore, readers can understand.
Answer 5: Full name of SSJ100, company designer and origin country were added.
Comment 6: Ansys Mechanical Software in line 82 requested to add license number, version of Ansys.
Answer 6: Information version of Ansys was added.
Comment 7: In same line 82 is Ansys software Ansys® or Ansys Trade mark? any of them add please.
Answer 7: “Ansys®” was added.
Comment 8: Figure 3 and Figure 4 can authors are what software used by others? if different from
Ansys requested to add.
Answer 8: For SSE calculations, parametric models created in Siemens Digital Industries Software® Siemens NX were used.
Comment 9: Is there any uncertainty in the work?
Answer 9: All uncertainties were corrected.
Comment 10: Figure can Authors represent the same graph in origin, please?
Answer 10: Figures were represented in better quality.
Reviewer 2 Report
This paper deal with the LCF of the structurally similar elements (SSE) instead of the titanium booster drum of gas turbine engines. Simulation of the responsible component were conducted in both perspectives, testing and simulation, and technical achievements seems to be well-prepared starting from simulation. However, some modification issues were found in first version as stated below.
[1] The verification of the constructed simulation model was not conducted via experimental consequences so that the simulation results may not be reliable.
[2] How do you control the temperature of the responsible specimen during fatigue testing?
[3] How to extend the proposed LCF method for other mechanical components?
[4] The measurement process of SSE in chapter 4 was too concisely stated to understand it.
[5] The measured voltages in Table 3 were recommended to replace them with physical ones.
Minor>
[1] Figure format was not identical.
Author Response
Reviewer 2
Comment 1: The verification of the constructed simulation model was not conducted via experimental consequences so that the simulation results may not be reliable.
Answer 1: The verification of the drum certification model was carried out during testing and operation on engines. SSE model was created and calculated like a verified drum certification model and verified by strain gauges and cyclic deflection measurements.
Comment 2: How do you control the temperature of the responsible specimen during fatigue testing?
Answer 2: The temperature of the test sample was controlled by a thermocouple attached in the critical zone. According to the readings, the temperature in the region of the critical zone did not exceed the temperature in the laboratory. Due to the complexity of mounting the thermocouple and the possible impact on SSC, no temperature control was performed for the test series.
Comment 3: How to extend the proposed LCF method for other mechanical components?
Answer 3: For other parts, development of the type of structural element, choice of a loading scheme, testing machine and equipment are required; the main criterion is the similarity of SSC to SSE and the full-size part in the critical zone.
Comment 4: The measurement process of SSE in chapter 4 was too concisely stated to understand it.
Answer 4: Description of measured dimensions and measurement tools were added.
Comment 5: The measured voltages in Table 3 were recommended to replace them with physical ones.
Answer 5: The physical magnitudes of stresses were replaced by relative magnitudes on the recommendation of the inspection services.
Round 2
Reviewer 2 Report
Most of reviewer's concerns were cleared in the revised version. So, I recommended this paper for publication.